# Hepatic Crtc2 controls whole body energy metabolism via a miR-34a-Fgf21 axis

Hye-Sook Han[1], Byeong Hun Choi[1], Jun Seok Kim[1], Geon Kang[1] & Seung-Hoi Koo[1]

Liver plays a crucial role in controlling energy homeostasis in mammals, although the exact mechanism by which it influences other peripheral tissues has yet to be addressed. Here we show that Creb regulates transcriptional co-activator (Crtc) 2 is a major regulator of whole-body energy metabolism. *Crtc2* liver-specific knockout lowers blood glucose levels with improved glucose and insulin tolerance. Liver-specific knockout mice display increased energy expenditure with smaller lipid droplets in adipose depots. Both plasma and hepatic Fgf21 levels are increased in *Crtc2* liver-specific knockout mice, as a result of the reduced miR-34a expression regulated by Creb/Crtc2 and the induction of Sirt1 and Pparα. Ectopic expression of miR-34a reverses the metabolic changes in knockout liver. We suggest that Creb/Crtc2 negatively regulates the Sirt1/Pparα/Fgf21 axis via the induction of miR-34a under diet-induced obesity and insulin-resistant conditions.

---

[1] Division of Life Sciences, Korea University, 145 Anam-Ro, Seongbuk-Gu, Seoul 02841, Korea. Correspondence and requests for materials should be addressed to S.-H.K. (email: koohoi@korea.ac.kr)

Under fasting conditions, liver plays a major role in maintaining energy homeostasis in part via activation of gluconeogenesis and ketogenesis to provide enough fuels for critical organs including brain and skeletal muscles[1, 2]. The activation of these processes is achieved in part via an increased transcription of genes encoding key rate-limiting enzymes in the pathway. For example, among various transcription factors, cAMP-mediated activation of cAMP response element binding protein (Creb) has been shown to be responsible for the increased hepatic gluconeogenesis, while free fatty acid-dependent induction of peroxisome proliferator-activated receptor (Ppar) α

functions as a key transcription factor for the activation of fatty acid β-oxidation and ketogenesis in the liver.

Fibroblast growth factor (Fgf) 21 belongs to a subfamily of Fgf proteins that functions as an endocrine hormone[3]. It is mainly produced in the liver upon starvation or ketogenic diet via the transcriptional control by Pparα, Foxo1, and Atf4[4–7]. Interestingly, carbohydrate ingestion also enhances expression of this factor in a carbohydrate response element binding protein (Chrebp)-dependent manner[8]. Besides in the liver, cold exposure or mitochondrial stress also induce expression of Fgf21 in adipocytes and skeletal muscles, mainly functioning in an autocrine

or paracrine manner[9–12]. Fgf21 has been shown to improve energy homeostasis by increasing hepatic fatty acid oxidation and ketogenesis in the liver, thermogenesis of brown adipose tissue (BAT), and browning of white adipose tissue (WAT), thereby increasing whole-body energy expenditure in mammals.

Previously, Creb co-activator Crtc2 was shown to be essential in the regulation of chronic activation of gluconeogenesis in the liver, and it was suggested that prolonged activation of Crtc2 under insulin resistance might be crucial for hyperglycemia in that setting[13–15]. However, the physiological importance of Crtc2 in the liver was not indisputably verified due to the lack of studies that utilized liver-specific *Crtc2* knockout mice[16, 17]. Given the recent report suggesting a role of Crtc2 in the regulation of lipid metabolism, which also utilized a systemic *Crtc2* knockout model, it is necessary to delineate the role of hepatic Crtc2 in the control of various metabolic pathways in more appropriate mouse models.

In this study, by utilizing the liver-specific *Crtc2* knockout mice, we uncovered a previously unidentified mechanism by which Crtc2 controls whole-body energy metabolism. We found that the depletion of hepatic *Crtc2* is beneficial in relieving not only hyperglycemia but also improving whole-body energy metabolism in diet-induced obesity (DIO) mouse models by promoting an Fgf21-dependent pathway.

## Results

**Hepatic depletion of *Crtc2* improves glucose homeostasis.** We generated liver-specific *Crtc2* knockout (*Crtc2^LKO*) mice to investigate the physiological role of this factor in energy metabolism. Albumin promoter-mediated expression of Cre recombinase effectively deleted *Crtc2* expression in the liver, but not in other tissues (Supplementary Fig. 1a–c). As expected, mice with hepatic depletion of *Crtc2* displayed reduced blood glucose levels and improved insulin tolerance and pyruvate tolerance compared with the control, with concomitant reduction in gluconeogenic genes such as phosphoenolpyruvate carboxykinase (*Pepck*) and glucose-6-phosphatase catalytic subunit (*G6pase*) in the liver of mice under normal chow diet (NCD) (Supplementary Fig. 1d–f). Interestingly, lipid droplets in the brown and WATs were smaller in *Crtc2^LKO* mice compared with the control (*Crtc2^f/f*) mice, while hepatic and plasma triglycerides (TG) were unchanged (Supplementary Fig. 2a, b). In spite of the improved insulin tolerance, we did not observe changes in the insulin signaling pathway as confirmed by the lack of changes in p-Akt levels between livers of *Crtc2^LKO* mice and those of *Crtc2^f/f* mice (Supplementary Fig. 2c).

To elaborate the potential function of hepatic Crtc2 in the obese setting, we fed mice with a high-fat diet (HFD) for 8 weeks. We observed a significant impact on glucose metabolism by liver-specific depletion of *Crtc2*, as evidenced by the improved glucose, pyruvate, glucagon, and insulin tolerance in *Crtc2^LKO* mice

compared with *Crtc2^f/f* mice in HFD conditions (Fig. 1a; Supplementary Fig. 3a). While *Crtc2^LKO* mice displayed reduced fasting hepatic glycogen levels compared with the control, hepatic glycogen levels were not different between the two genotypes during feeding (Supplementary Fig. 3b). In line with the improved insulin tolerance, the peripheral insulin signaling might be more active in *Crtc2^LKO* mice in comparison to *Crtc2^f/f* mice, as shown by the increased tyrosine phosphorylation of insulin receptor (IR) and the increased serine phosphorylation of Akt in response to insulin in the liver or visceral WAT (Supplementary Fig. 3c, d). Blood glucose and insulin levels were also lower in *Crtc2^LKO* mice, corroborating the suggested role of Crtc2 in glucose metabolism (Fig. 1b).

**Hepatic *Crtc2* deficiency enhances energy expenditure.** Unexpectedly, we found that both plasma and hepatic triacylglycerol levels were considerably lower in *Crtc2^LKO* mice compared with *Crtc2^f/f* mice (Fig. 1c), arguing against a recent report showing an inhibitory role of Crtc2 on hepatic lipogenesis, a study that utilized systemic *Crtc2* knockout mice[18]. In fact, we observed rather reduction in lipogenic gene expression in livers of *Crtc2^LKO* mice in comparison to those of *Crtc2^f/f* mice, suggesting that effects of liver-specific depletion of *Crtc2* on hepatic lipogenesis are different from those of systemic *Crtc2* knockout mice, perhaps due in part to the improved insulin sensitivity in *Crtc2^LKO* mice (Supplementary Fig. 3e). In line with the data with the improved lipid metabolism upon chronic depletion of *Crtc2* in the liver, we observed reduced size in lipid droplets in the liver, BAT, subcutaneous white adipose tissue (scWAT), and visceral white adipose tissue (visWAT) (Fig. 1d, e). Furthermore, we observed that the fat body mass was also reduced in *Crtc2^LKO* mice compared with *Crtc2^f/f* mice (Supplementary Fig. 3f). Reduced lipid droplet size in the liver may suggest the increased degradation of lipid droplets by autophagy (or lipophagy) in the *Crtc2^LKO* liver compared with the *Crtc2^f/f* liver[19]. However, we did not observe consistent changes in expression of key components of autophagy between livers of *Crtc2^LKO* mice and those of *Crtc2^f/f* mice (i.e., reduced Atg7 expression and induced Atg12 expression, together with no changes in p62 levels or LC3-II levels), arguing against the potential involvement of lipophagy in this setting (Supplementary Fig. 4a, b). As the reduced lipid droplets in fat cells and reduced body weight gain upon HFD in *Crtc2^LKO* mice may indicate an increased energy expenditure (Fig. 1f), we performed an indirect calorimetric assay by using mice under HFD conditions. Indeed, we observed a small but significant increase in energy expenditure from *Crtc2^LKO* mice compared with *Crtc2^f/f* mice, which could lead to the reduced body weight for the mutant group, without thereby changes in food intake, water consumption, or locomotor activity (Fig. 1g; Supplementary Fig. 4c, d).

**Fig. 1** Hepatic Crtc2 controls systemic glucose and lipid metabolism in mammals. **a** Glucose tolerance test (GTT, left), pyruvate tolerance test (PTT, middle), and insulin tolerance test (ITT, right) showing effects of chronic depletion of hepatic *Crtc2* in mice under HFD for 8 weeks on glucose metabolism and insulin signaling (n = 13 for *Crtc2^f/f* mice and n = 11 for *Crtc2^LKO* mice). Area under the curve (AUC) for each test was also shown. **b** Ad libitum (fed) or 16 h-fasting (fasted) blood glucose levels (left), and ad libitum (fed) or 16 h-fasting (fasted) insulin levels (right) from either *Crtc2^f/f* mice or *Crtc2^LKO* mice under HFD for 9 weeks (n = 13 for *Crtc2^f/f* mice and n = 11 for *Crtc2^LKO* mice). **c** Plasma triglycerides (TG) levels (left) and hepatic TG levels (right) from either *Crtc2^f/f* mice or *Crtc2^LKO* mice under ad libitum (fed) or 16 h-fasting (fasted) conditions under HFD for 9 weeks (n = 13 for *Crtc2^f/f* mice and n = 11 for *Crtc2^LKO* mice). **d** Paraffin-embedded sections of liver tissues from 8-week HFD-fed mice were stained with hematoxylin and eosin (H&E, left). Frozen liver sections were also stained with oil red O (right). Data represent three independent experiments (n = 3 mice per group). **e** Paraffin-embedded sections of subcutaneous white adipose tissues (scWAT, left), visceral white adipose tissues (visWAT, middle), and brown adipose tissues (BAT, right) from 16 h-fasted, 9-week HFD-fed *Crtc2^f/f* mice or *Crtc2^LKO* mice were stained with H&E. Data represent three independent experiments (n = 3 mice per group). **f** Body weight from either *Crtc2^f/f* mice or *Crtc2^LKO* mice under HFD (n = 13 for *Crtc2^f/f* mice and n = 11 for *Crtc2^LKO* mice). **g** Energy expenditure (top and bottom left for quantitation) and locomotor activity (bottom right) were measured from 7-week HFD-fed *Crtc2^f/f* mice or *Crtc2^LKO* mice by using metabolic cage (n = 10 for *Crtc2^f/f* mice and n = 9 for *Crtc2^LKO* mice). Note that the HFD was initiated at 4 weeks of age and maintained throughout the experimental period. Data in **a**, **b**, **c**, **f** and **g** represent mean ± s.e.m. (*P < 0.05, **P < 0.01, t-test **a–c** and **g** or Tukey–Kramer multiple comparisons **f**)

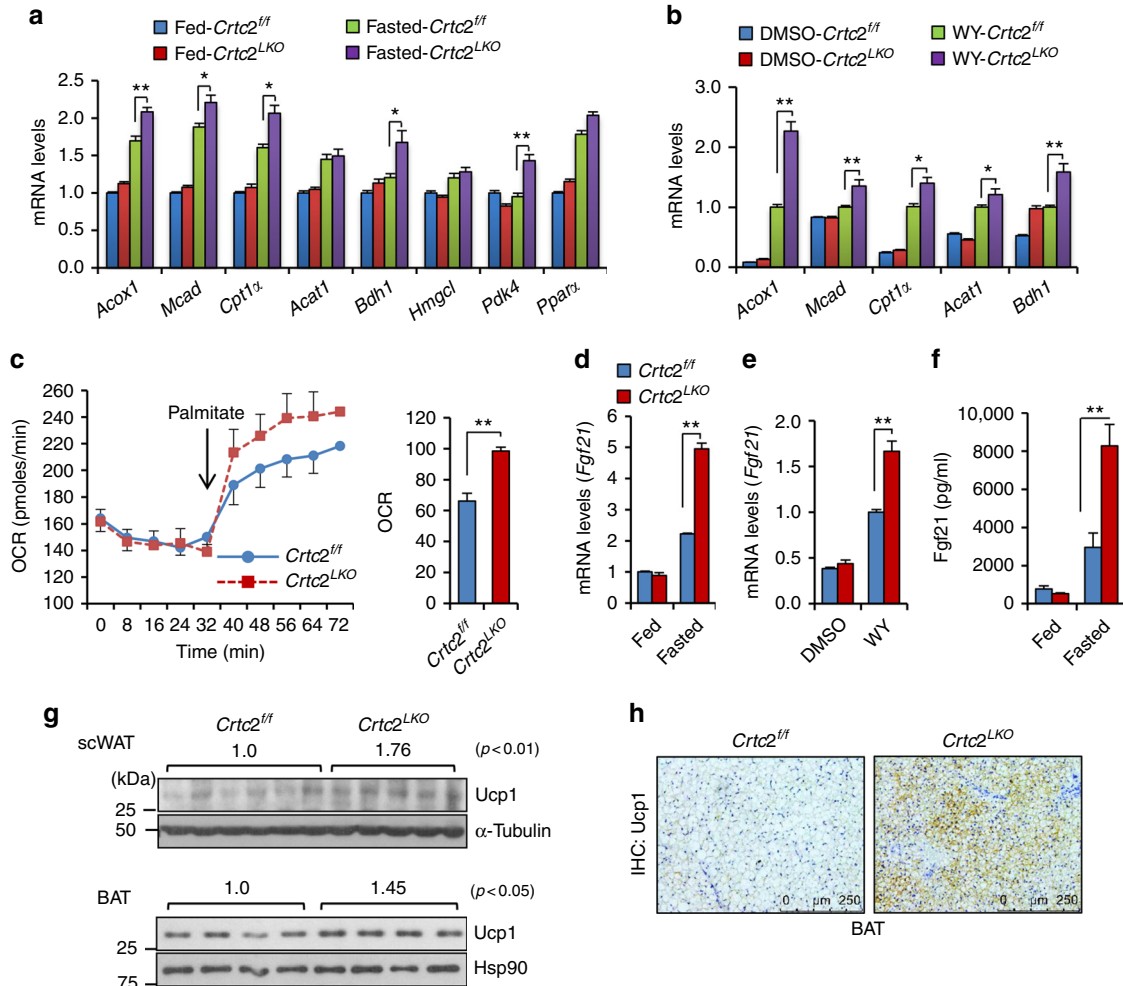

**Fig. 2** Chronic depletion of hepatic *Crtc2* promotes a Pparα-dependent transcriptional program in the liver. **a** Effects of chronic depletion of hepatic *Crtc2* in mice under HFD for 9 weeks on fatty acid beta oxidation genes (Q-PCR, *n* = 5–7 mice per group). **b** Effects of *Crtc2* knockout on fatty acid beta oxidation genes in primary hepatocytes in the absence or in the presence of Pparα agonist WY14643 (10 μM for 16 h) (Q-PCR, *n* = 3 sets of cells per group). **c** Oxygen consumption rate was measured from primary hepatocytes of *Crtc2^{f/f}* mice or *Crtc2^{LKO}* mice as a surrogate for the rate of fatty acid oxidation (*n* = 5 sets of cells per group). Area under the curve was also shown at right. **d** Effects of chronic depletion of hepatic *Crtc2* in mice under HFD for 9 weeks on *Fgf21* expression (Q-PCR, *n* = 5–7 mice per group). **e** Effects of *Crtc2* knockout on *Fgf21* expression in primary hepatocytes in the absence or in the presence of Pparα agonist WY14643 (10 μM for 16 h) (Q-PCR, *n* = 3 sets of cells per group). **f** Effects of chronic depletion of hepatic *Crtc2* on plasma Fgf21 levels in 16 h-fasted mice under HFD for 8 weeks (*n* = 7–11 mice per group). **g** Effects of hepatic *Crtc2* knockout on Ucp1 protein levels in the subcutaneous WAT from 6 h-fasted mice under HFD for 8 weeks (top). Quantitation of Ucp1 levels of each group was shown above the Ucp1 bands (*P* < 0.01, *t*-test). Effects of hepatic *Crtc2* knockout on Ucp1 protein levels in the BAT from 6 h-fasted mice under HFD for 8 weeks (bottom). Quantitation of Ucp1 levels of each group was shown above the Ucp1 bands (*P* < 0.05, *t*-test). **h** Immunohistochemistry analysis showing effects of chronic liver-specific depletion of *Crtc2* in mice on Ucp1 expression in BAT. Data represent four independent experiments (*n* = 4 mice per group). Note that the HFD was initiated at 4 weeks of age and maintained throughout the experimental period. Data in **a**–**e** and **g** represent mean ± s.d. (*\*P* < 0.05, *\*\*P* < 0.01, *t*-test (**c**) or Tukey–Kramer multiple comparisons **a**, **b**, **d** and **e**), and data in **f** represent mean ± s.e.m. (*\*\*P* < 0.01, Tukey–Kramer multiple comparisons)

**Depletion of hepatic *Crtc2* promotes a Pparα-dependent pathway.** Since Crtc2 is a transcriptional co-activator, we reasoned that the impact of hepatic Crtc2 depletion on energy metabolism should be stemmed from changes in gene expression. Thus, we performed an extensive quantitative RT-PCR (Q-PCR) analysis to detect potential target genes of Creb/Crtc2 that could be causal to the current metabolic phenotype shown in this study. We noticed that genes that code for enzymes in the fatty acid oxidation and ketogenesis were generally increased in livers of *Crtc2^{LKO}* mice in comparison to those of *Crtc2^{f/f}* mice (Fig. 2a). In compliance with this data, we observed higher plasma ketone body levels in *Crtc2^{LKO}* mice, and the increased expression of fatty acid oxidation genes together with the increased oxygen consumption in response to palmitate in primary hepatocytes

from *Crtc2^{LKO}* mice compared with *Crtc2^{f/f}* mice (Fig. 2b, c; Supplementary Fig. 4e). Specifically, the expression of *Fgf21*, a hepatokine that was shown to improve whole-body energy metabolism by accelerating lipid usage and non-shivering thermogenesis in adipocytes[11, 20], was induced more than two-fold in livers of *Crtc2^{LKO}* mice in comparison to those of *Crtc2^{f/f}* mice during fasting conditions, which was confirmed in isolated primary hepatocytes (Fig. 2d, e). Plasma Fgf21 levels were also higher in *Crtc2^{LKO}* mice compared with *Crtc2^{f/f}* mice under fasting (Fig. 2f). Indeed, we observed increased Ucp1 protein levels in BAT and subcutaneous WAT, suggesting that higher plasma concentration of Fgf21 in *Crtc2^{LKO}* mice greatly affected lipid metabolism in adipocytes (Fig. 2g, h). Unexpectedly, expression of gluconeogenic genes was largely unaltered by

chronic depletion of *Crtc2* in the liver under HFD, although the primary hepatocytes from *Crtc2^LKO* mice showed a dampened transcriptional response to forskolin (Supplementary Fig. 5a, b). We noticed a potential compensatory increase in messenger RNA (mRNA) levels of *C/ebpβ* (Supplementary Fig. 5c), a transcription factor that was also shown to enhance transcription of

gluconeogenic genes[21]. In addition, higher plasma cortisol levels in *Crtc2^LKO* mice may ensure the normal expression levels of gluconeogenic genes even in the absence of *Crtc2* (Supplementary Fig. 5d), as suggested by the recent report showing that Fgf21 can induce hepatic gluconeogenesis by inducing secretion of cortisol from the adrenal gland in vivo[22].

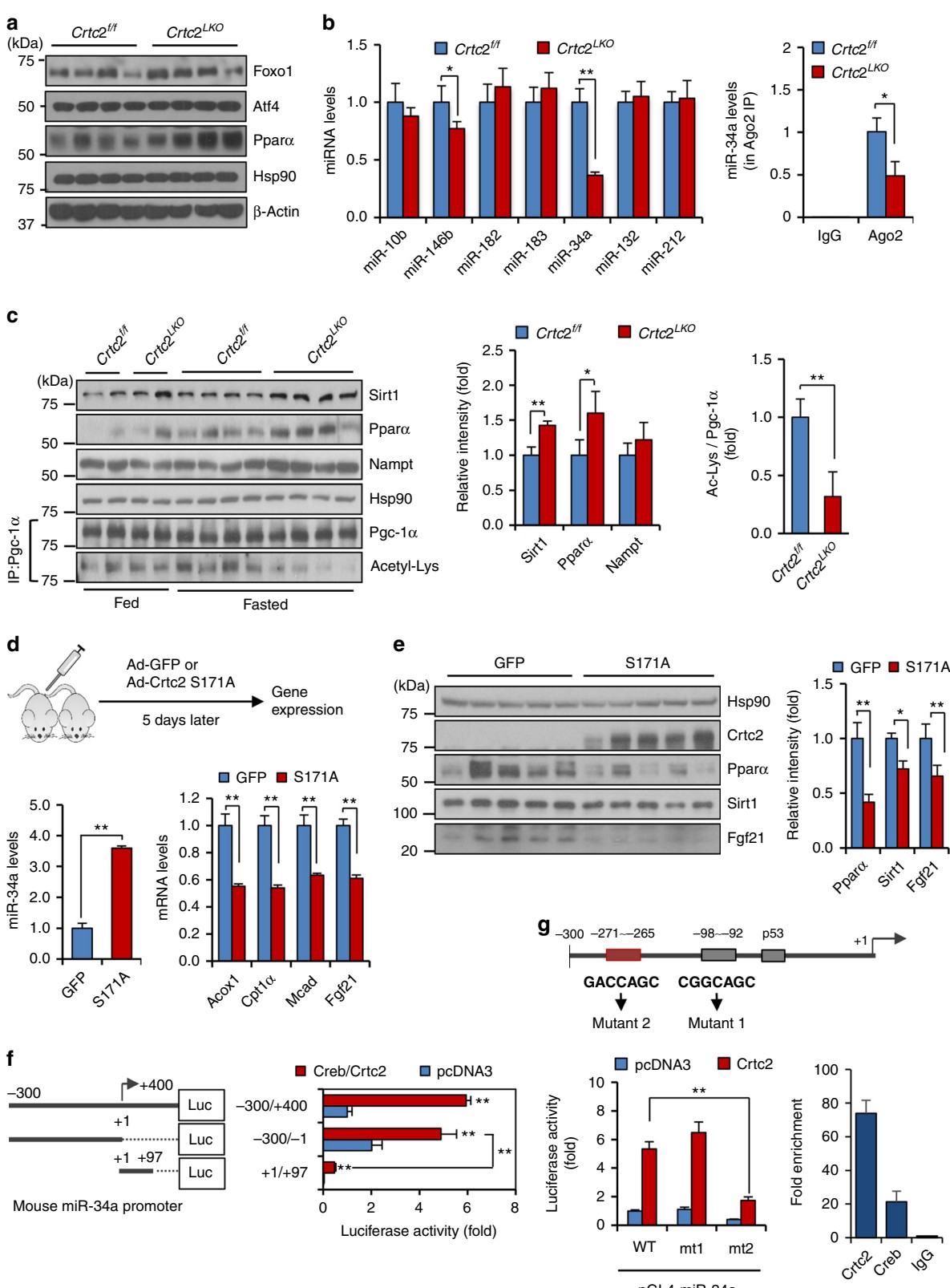

**Crtc2 regulates Sirt1–Pparα–Fgf21 axis by controlling miR-34a**. Various transcription factors such as Pparα, Atf4, and Foxo1 were shown to be directly involved in the regulation of Fgf21 expression[4–7]. While neither Atf4 nor Foxo1 levels were altered, Pparα protein levels were higher in the livers of *Crtc2^LKO* mice in comparison to those of *Crtc2^f/f* mice under fasting conditions (Fig. 3a). In addition, we did not observe any changes in hepatic Chrebp expression between the two groups of mice (data not shown). These data suggest that Crtc2 could directly control expression of Pparα at the protein levels, perhaps through a direct regulation of microRNAs. Thus, we performed small RNA sequencing analysis by using livers of either *Crtc2^LKO* mice or *Crtc2^f/f* mice under HFD. We noticed a significant change in the expression of several microRNAs in livers of *Crtc2^LKO* mice including miR-34a, miR-10b, and miR-146b (Supplementary Table 1). Among the potential microRNAs, only miR-34a level was confirmed to be reduced more than two-fold upon depletion of Crtc2 by Q-PCR (Fig. 3b). Primary miR-34a levels were also reduced in the livers of *Crtc2^LKO* mice in comparison to those of *Crtc2^f/f* mice under HFD-fed conditions, corroborating our hypothesis that the transcription of miR-34a requires the Creb/Crtc2 transcriptional machinery (Supplementary Fig. 5e). As shown before, miR-34a level was higher in the livers of HFD-fed mice compared with the control (Supplementary Fig. 5f)[23–25]. Interestingly, previous reports suggested that miR-34a reduces expression of various targets such as Sirt1, Pparα, and Nampt in the liver, which could directly affect activity of Pparα[23, 24, 26]. Indeed, we observed that protein levels of Sirt1, Pparα, and Nampt were higher in livers of *Crtc2^LKO* mice in comparison to those of *Crtc2^f/f* mice (Fig. 3c). Furthermore, we observed the decreased acetylation of peroxisome proliferator activating receptor gamma (Pgc)-1α, a known co-activator for Pparα, presumably due to the increased expression of Sirt1 and Nampt proteins. These data corroborate our hypothesis that reduced expression of miR-34a resulted in the activation of Pparα-transcriptional machinery, leading to the enhanced expression of Fgf21 in the *Crtc2^LKO* liver during fasting. Overexpression of miR-34a in the liver resulted in the decreased expression of Pparα and Fgf21 as well as Pparα target genes, confirming that miR-34a directly controls Fgf21 expression via a Pparα-dependent transcriptional pathway (Supplementary Fig. 6a–c).

We did not observe significant changes in miR-34a expression either by fasting or the liver-specific depletion of Crtc2 under NCD conditions (data not shown). We reasoned that Creb/Crtc2 transcriptional machinery per se might not be sufficient to enhance transcription of miR-34a but could be responsible for the increased expression of hepatic miR-34a in the DIO conditions, in which conditions a consistent activation of Crtc2 was observed[14]. To mimic the DIO conditions that results in the consistent activation of Crtc2, we utilized adenovirus expressing Crtc2 S171A, a constitutively active form of Crtc2[15].

Overexpression of Crtc2 S171A resulted in the increased expression of miR-34a in the liver, with concomitant decrease in protein levels of Pparα, Sirt1, and Fgf21, together with Pparα target genes (Fig. 3d, e; Supplementary Fig. 6d). Luciferase reporter assay also showed that promoter activity of miR-34a was induced by co-expression of Creb and Crtc2 in cultured cells (Fig. 3f; Supplementary Fig. 7a). p53 has been listed as a major transcriptional regulator for miR-34a transcription[27, 28]. It is of note that Crtc2 did not further activate p53-dependent increase in miR-34a promoter activity, showing that these two pathways do not converge on the transcriptional regulation of miR-34a (Supplementary Fig. 7b). Promoter sequences for miR-34a were previously delineated, and we identified several potential cAMP response elements (CRE). The mutagenesis study revealed that among the potential CREs, the element at −132 for the human promoter and the one at −271 for the mouse promoter were critical in Creb/Crtc2 transcriptional response (Fig. 3g; Supplementary Fig. 7c). The result was also confirmed by chromatin immunoprecipitation assay both in HepG2 cells and mouse primary hepatocytes. These data collectively suggest that Creb/Crtc2 is critical in the regulation of Pparα-dependent lipid metabolism not only in the liver but also at the whole-body level by controlling hepatic expression of Fgf21 in the DIO conditions.

**Depletion of Fgf21 reverts the effect of Crtc2 knockout in DIO mice**. We then wanted to confirm whether the effect of liver-specific depletion of Crtc2 on the whole-body energy metabolism is due to the elevation in Fgf21 levels. To this end, we generated Fgf21 liver-specific mice (*Fgf21^LKO* mice) together with Crtc2/Fgf21 double liver-specific knockout mice (*Crtc2/Fgf21^LKO* mice) by using albumin-Cre transgene. While single knockout of Fgf21 in mice almost completely diminished the plasma Fgf21 levels compared with WT mice, we failed to observe any significant changes in either blood glucose levels or body weight between the two genotypes. Similarly, we did not observe any differences in body weight or lipid droplet size in the liver, scWAT, visWAT, and BAT between *Crtc2^f/f* mice and *Fgf21^LKO* mice, suggesting that the simple depletion of hepatic Fgf21 per se might not provoke metabolic phenotypes under DIO conditions (Supplementary Fig. 8a–e). On the other hand, concomitant depletion of Crtc2 and Fgf21 in the liver, which also almost completely diminished the plasma Fgf21 levels, reversed the beneficial effect of liver-specific depletion of Crtc2 in blood glucose levels, body weight, and whole-body lipid metabolism and adiposity, as we observed the increased lipid droplet size in liver, WAT, and BAT of *Crtc2/Fgf21^LKO* mice compared with the *Crtc2^LKO* mice (Supplementary Fig. 8a–e). In addition, Ucp1 protein levels (BAT) were increased by single knockout of Crtc2 in the liver, which were reduced by double knockout of Crtc2 and Fgf21 in the liver of mice (Supplementary Fig. 8f). Again, a single depletion of

**Fig. 3** Hepatic miR-34a is critical in reducing Sirt1-Pparα-mediated expression of *Fgf21* in a Creb/Crtc2-dependent manner. **a** Effects of hepatic *Crtc2* knockout on protein levels of Atf4, Foxo1, and Pparα in the livers of 16 h-fasted, 9-week HFD-fed mice. **b** Effects of hepatic *Crtc2* knockout on mature microRNA expression in the livers of 16 h-fasted, 9-week HFD-fed mice (left) (Q-PCR, $n = 4$ mice per group). Levels of AGO2-associated miR-34a were also shown (right) (Q-PCR, $n = 4$ mice per group). **c** Effects of hepatic *Crtc2* knockout on protein levels in the livers of 16 h fasted, 9-week HFD-fed mice. Quantitation of protein levels of each condition was shown at right. **d, e** Effects of expression of constitutively active form of Crtc2 (S171A) on hepatic genes. 8-week old C57BL/6 mice were injected with adenovirus for 5 days before being killed after 24 h fasting. Hepatic miR-34a levels (**d**, left), hepatic expression of Pparα target genes (**d**, right), hepatic protein levels (**e**, left), and quantitation of protein levels (**e**, right) from C57BL/6 mice that were infected with Ad-GFP or Ad-Crtc2 S171A adenovirus ($n = 5$ mice per group). **f** 5′- and 3′-deletion analysis was performed to map the putative Creb/Crtc2 response element on the promoter of mouse miR-34a (top). Luciferase reporter assay was performed in 293T cells to determine the effects of Creb/Crtc2 on miR-34a promoter activity (bottom). $N = 3$ independent experiments in triplicate. **g** Location of putative CREs on the mouse miR-34a promoter (top), the effects of CRE mutations on the mouse miR-34a promoter activity (left), and the chromatin immunoprecipitation assay showing the occupancy of Creb/Crtc2 over mouse miR-34a promoter (right) were shown. $N = 3$ independent experiments in triplicate. Note that the HFD was initiated at 4 weeks of age and maintained throughout the experimental period (**a–c**). Data in **b–g** represent mean ± s.d. (*$P < 0.05$, **$P < 0.01$, $t$-test)

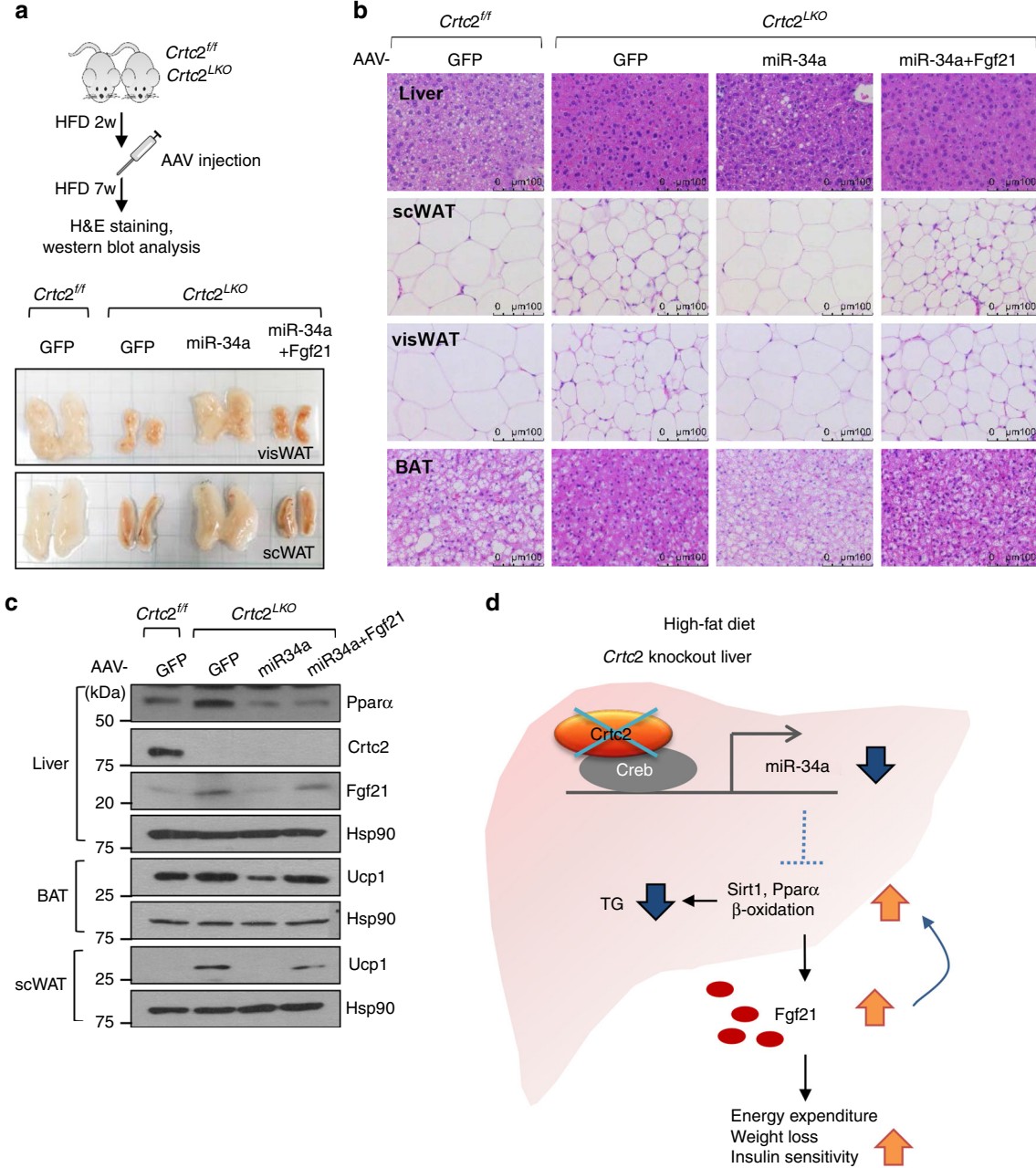

**Fig. 4** Effect of miR-34a on adiposity is reversed by co-expression of Fgf21 in *Crtc2* liver-specific knockout mice. **a**–**d** Effects of miR-34a and/or Fgf21 on *Crtc2^LKO* mice under 9-week HFD (*n* = 5 mice per group). Schematics of AAV injection (**a**, top) and the effect of miR-34a and Fgf21 on fat size (**a**, bottom) were shown. H&E staining of the liver, scWAT, visWAT, BAT of each group of mice were shown (**b**). **c** Western blot analysis showing hepatic protein levels, as well as Ucp1 protein levels in scWAT and BAT of each genotype. Data represent four independent experiments (*n* = 4 mice per group). Note that the HFD was initiated at 4 weeks of age and maintained throughout the experimental period. **d** Schematic diagram showing the effects of Creb/Crtc2-driven expression of miR-34a on hepatic fatty acid metabolism and whole-body energy homeostasis under diet-induced obesity (DIO) and insulin-resistant conditions. Depletion of *Crtc2* promotes the activation of Sirt1/Pparα pathway under DIO conditions, leading to the increased hepatic fatty acid beta oxidation and the amelioration of fatty liver symptoms. In addition, enhanced hepatic Fgf21 expression is also essential in improving lipid metabolism in the peripheral tissues including the liver and adipose tissues, resulting in the increased energy expenditure and the improved insulin sensitivity

hepatic *Fgf21* only did not affect BAT Ucp1 levels in DIO conditions. These data suggest that the effect of liver-specific depletion of *Crtc2* on the lipid homeostasis and the energy metabolism might be mainly functioning via the increased plasma Fgf21. Knockout of *Fgf21* in the liver effectively removed the beneficial effect of the increased Fgf21 in this setting, reversed the effect of *Crtc2* depletion on hepatic lipid contents and overall adiposity under DIO conditions. These data suggest that increased plasma Fgf21 under fasting could promote improved lipid homeostasis

while the depletion of hepatic *Fgf21* (and thus reducing plasma Fgf21 levels) per se might not instigate metabolic disorders.

**Effect of miR-34a on adiposity is blocked by co-expression of Fgf21.** To test whether increased Pparα/Fgf21 program under *Crtc2* depletion is indeed through miR-34a-dependent pathway, we expressed miR-34a or miR-34a plus Fgf21 in the liver of *Crtc2^LKO* mice via adeno-associated virus-mediated gene delivery.

As shown before, plasma Fgf21 levels were increased in *Crtc2^LKO* mice in comparison to the control, which were reduced significantly by re-expression of miR-34a. As expected, co-expression of Fgf21 together with miR-34a restored higher plasma Fgf21 levels (Supplementary Fig. 9a). Body weight, as well as size of WATs, was reduced in *Crtc2^LKO* mice compared with the control. Re-expression of miR-34a reversed this phenotype, and co-expression of Fgf21 inhibited the effect of miR-34a (Fig. 4a; Supplementary Fig. 9b–d). The changes in lipid homeostasis were indeed prominent as shown by hematoxylin and eosin (H&E) staining data of liver and adipose tissues (Fig. 4b). MiR-34a effectively blocked the changes in lipid droplet size of liver, WAT, and BAT in hepatic *Crtc2* depleted mice, and co-expression of Fgf21 also interfered with the effect of miR-34a on adiposity in mice. Mechanistically, the levels of Pparα protein and its target Fgf21 protein were induced in livers of *Crtc2^LKO* mice in comparison to those of *Crtc2^f/f* mice. Re-expression of miR-34a in *Crtc2^LKO* led to the reduction of both proteins, while co-expression of miR-34a and Fgf21 restored hepatic Fgf21 protein levels but not Pparα protein levels. As well, increased Ucp1 protein levels in BAT and scWAT of *Crtc2^LKO* mice were reduced by miR-34a expression, and they were restored by co-expression of miR-34a and Fgf21, confirming our hypothesis that Crtc2-miR-34a axis interferes with Pparα-Fgf21 pathway under HFD-induced obesity and insulin-resistant conditions (Fig. 4c).

## Discussion

In this study, we identified a novel mechanism by which Crtc2 controls whole-body energy homeostasis via controlling expression of hepatokine Fgf21. Fgf21 has been shown to improve energy homeostasis in mammals by acting directly on the central nervous systems as well as peripheral tissues including adipose tissues[20]. These studies were corroborated by use of tissue-specific βKlotho knockout mouse models, a co-receptor for Fgf19 and 21 that is crucial in mediating intracellular hormonal signals[29, 30]. Interestingly, a recent study disputed the role of βKlotho in mediating Fgf21 response in WATs, suggesting that further study is necessary to clarify the molecular characteristics of plasma membrane receptors of Fgf21 in tissue-specific manners[31]. Although it was shown that Fgf21 improves hepatic lipid metabolism by enhancing fatty acid oxidation and ketogenesis and reducing lipogenesis, the effects of Fgf21 on the liver may be indirect via the action of sympathetic nerve activity[30, 32]. On the other hand, a recent report showed a role of direct action of Fgf21 on hepatic metabolism, suggesting that the issue is yet to be clearly determined[33].

It is noteworthy to mention that the effect of Crtc2 depletion on Fgf21 levels was only prominent under fasting conditions (Fig. 2f). Being regarded as an endocrine hormone, this type of regulation of Fgf21 levels is not unprecedented. Indeed, most endocrine hormones are regulated in response to the nutrient status. Examples include insulin (during feeding), glucagon (during fasting), and many incretins such as GLP-1 and GIP-1 (during feeding), which mainly function in the specific nutrient status and countered by counter-regulatory hormones upon changes in nutrient state. While the secretion of such hormones are regulated at the level of exocytosis, there are examples of endocrine hormones such as GLP-1 that are also regulated at the level of transcription. In addition, counter-regulatory hormones exist to limit the action of certain hormones for a prolonged period of time. *Crtc2* liver-specific knockout mice showed increased insulin sensitivity, thus we suspected that insulin could counteract to reduce Fgf21 levels during feeding by reducing transcriptional activation of hepatic *Fgf21* via the inhibition of Pparα activity.

Based on our data, we would like to propose that the tight regulation of hepatic Creb/Crtc2 function is critical in the maintenance of energy homeostasis in mammals. Under hepatic insulin resistant state, prolonged activation of Crtc2 is not only responsible for the increased glucose production from the liver[14], but also critical in promoting the accumulation of lipids in the liver and peripheral tissues in part by activating miR-34a-mediated repression of Sirt1/Pparα activity and *Fgf21* expression (Fig. 4d). As noted before, Fgf21 can function to promote hepatic fatty acid oxidation and induce adaptive thermogenesis in adipocytes, leading to the decreased peripheral fat mass, increased energy expenditure, and improved insulin sensitivity[11, 33]. Thus, the inhibition of miR-34a by reducing Crtc2 activity in the liver could be beneficial in combating obesity and insulin resistance by promoting whole-body energy metabolism in an Fgf21-dependent manner.

## Methods

**Animal experiments**. *Crtc2* conditional knockout mice were generated by Macrogen (Seoul, Korea) by using *Crtc2* mutant ES cell clone (#EPD0197_3_A08) from EUCOMM (European conditional mouse consortium). The knockout first line was first mated with CAG-Flp transgenic mice (Jackson lab) to discard the FRT sites, resulting in the generation of *Crtc2* flox/flox mice that contain the loxP sites surrounding the critical exon 4 of *Crtc2* gene. *Crtc2* flox/flox mice were then crossed with albumin-Cre transgenic mice (Jackson lab) to generate liver-specific *Crtc2* knockout mice. *Fgf21* flox/flox mice were purchased from the Jackson lab. *Fgf21* flox/flox mice were crossed with *Crtc2^LKO* mice to produce *Crtc2/Fgf21^LKO* mice. All mouse lines were backcrossed with C57BL/6 for at least five times before being used for the experiment.

For induction of obesity and insulin resistance, male 4-week-old mice were fed a HFD (60% fat diet, D12492; Research Diets) for 3–12 weeks (specific conditions are shown in the figure legends). For the glucose tolerance test (GTT) or pyruvate tolerance test (PTT), 16 h-fasted mice were intraperitoneally injected with a bolus of glucose or pyruvate (2 g/kg of body weight for NCD and 1.5 g/kg for HFD). For the insulin tolerance test (ITT) or the glucagon tolerance test, 6 h-fasted mice were intraperitoneally injected with a bolus of insulin (0.75 units of insulin/kg body weight) or glucagon (15 μg of glucagon/kg body weight). Blood glucose was measured from tail vein blood with an automatic glucose monitor (One Touch, LifeScan). For insulin signaling, either PBS or insulin (0.1 units per mouse) was injected intraperitoneally for 10 min, and the liver and the WAT were collected for the further analyses. Neither randomization nor blinding was used for the animal experiments in the study. Sample size was determined using previous experimental studies for metabolic assessment. All procedures were performed in a specific pathogen-free facility at the Gyerim Experimental Animal Resource Center, Korea University (12:12 h light–dark cycle, maintained at 22 °C), based on the protocols that were approved by the Korea University Institutional Animal Care and Use Committee.

**Culture of primary hepatocytes**. Mouse primary hepatocytes from 8- to 10-week-old C57BL/6N mice were isolated by collagenase perfusion methods[34]. After the isolation, the perfused liver was minced and was subject to the percoll gradient to selectively purify hepatocytes. Cells were then maintained in medium 199 (M199; Sigma-Aldrich, St. Louis, MO) with 10% fetal bovine serum (FBS; HyClone, Logan, UT), 100 U/ml penicillin, 100 μg/ml streptomycin, and 10 nM of dexamethasone. After the attachment, cells were then treated with various reagents as well as adenoviruses as shown in the figure legends.

**Western blot analysis and immunoprecipitation**. Western blot analyses were performed by using 10–60 μg of protein extracts as described previously[35]. Antibodies against Crtc2 and phosphor-Tyr IR were purchased from Calbiochem, an antibody against Crtc3 was from Bethyl lab, antibodies against Akt, phosphor Akt (T308, S473), Foxo1, acetyl lysine, autophagy proteins were from Cell Signaling, antibodies against β-actin and α-tubulin were from Sigma, an antibody against Sirt1 was from Millipore, antibodies against Hsp90, Atf4, and IRβ were from Santa Cruz, an antibody against Nampt was from Enzo life sciences, antibodies against Pparα and Ucp1 were from Abcam, an antibody against Fgf21 was from R&D systems. Acetylation of Pgc-1α was determined by the immunoprecipitation with Pgc-1α antibody[36], followed by the detection of acetylated lysine residues with an antibody against acetyl lysine. Band density was quantified by using the Image J software (NIH). Specific conditions for antibodies used in the western blot analysis are summarized in Supplementary Table 2.

**Histological analysis**. Liver and fat tissues were isolated from mice and were fixed with 10% formalin (Sigma). Histological changes were examined by H&E stain. Neutral lipid accumulation in liver was analyzed by Oil red O (ORO) staining

(Sigma) on the frozen liver tissues. Mayer's hematoxylin was used as counter-staining, respectively. Slides were observed with a light microscope (Leica DMi 8).

**Plasmids.** Human miR-34a promoter was excised from the pGL3-PMT-miR-34a plasmid (a gift from Judy Lieberman (Addgene plasmid #25799)[37], and was sub-cloned into the pGL4 vector to generate the human full-length miR-34a luciferase plasmid. For the mouse miR-34a promoter analysis, genomic sequences containing the miR-34a promoter were amplified by PCR from the mouse genomic DNA (Promega G309a), and were cloned into the pGL4 vector. Mutagenesis on the CREB binding sites within the promoter was performed by using the QuikChange Site-Directed Mutagenesis Kit according to the manufacturer's protocol (Strata-gene). All constructs were confirmed by sequencing.

**Luciferase reporter assays.** 293T cells were purchased directly from ATCC, and were routinely checked for the mycoplasma contamination. Cells were plated onto 24-well plates and then maintained in DMEM (Hyclone) supplemented with 10% FBS, penicillin (100 U/ml), and streptomycin (100 μg/ml). Each transfection was performed with 100 ng of pGL4-h/m miR-34a promoter, 50 ng of β-galactosidase, 50 ng of expression vector for Creb or Crtc2 by using TransIT-LT1 reagent (Mirus) for 48 h. Experiments were performed in triplicates and repeated at least three times. Luciferase activity was determined with the Promega Luciferase Assay Kit according to the manufacturer's protocol and normalized by β-galactosidase activity.

**RNA extraction and quantitative PCR.** Total RNA was isolated by using Trisure (Bioline) and RNeasy Kit (Qiagen) according to the manufacturer's protocols. Complimentary DNA was synthesized by using Goscript Reverse transcription systems (Promega) as described in the manufacturer's protocol. Quantitative real-time PCR (Q-PCR) analysis was performed in triplicate by using a SensiFAST SYBR green mix (Bioline) and CFX connect real-time system (Bio-Rad). The mRNA levels were normalized to ribosomal L32 or Gapdh. For microRNA (miRNA) quantification, total RNA was reverse transcribed by using the miScript II RT Kit (Qiagen). Primers specific for mouse miR-34a, miR-10b, miR-146b, miR-212, miR-132, and miR-582 were purchased from Qiagen. The values were normalized to RNU6B (Qiagen). Sequences for gene-specific primers are shown in Supplementary Table 3.

**Chromatin immunoprecipitation assay.** Mouse primary hepatocytes or HepG2 were plated in 100 mm dish. After treatment with 100 nM glucagon or 10 μM Forskolin for 2 h, cells were cross-linked with 1% formaldehyde for 10 min at 37 °C and stopped by the addition of glycine to a final concentration of 0.125 M. The ChIP assay was performed by using the ChIP Assay Kit (Millipore) according to the manufacturer's protocol as described previously. For human miR-34a pro-moter, primers −140(F): 5′-GATCCCGGGCTGGAGAGA-3′, and +120(R): 5′-CTGAGAAACACAAGCGTTTACCT-3′ were used. For mouse miR-34a promoter, primers −300(F): 5′-CTCCCTATTCCCCGCCTG-3′, and-1(R): 5′-CCCCCA ATCTGTGCAGTTAC-3′ were used. Total input was used as a normalization control.

**Serum FGF21, corticosterone, insulin level.** For detecting plasma levels of spe-cific hormones, Corticosterone ELISA Kit (ab108821), Fgf21 Kit (R&D systems MF2100), and Insulin Kit (Alpco) were used according to the manufacturer's protocols.

**Triglycerides level.** Total liver lipids were extracted according to the Folch method with a slight modification[38]. Briefly, mouse liver was homogenized with chloroform/methanol solution (2:1, v/v), and was centrifuged at room temperature. The supernatant was then washed with 1/5 volumes of 0.9% NaCl and was again centrifuged. After discarding the upper phase, the remaining solution was evapo-rated under vacuum. Hepatic and plasma TG contents were measured using an Enzymatic Colorimetric Assay Kit (Wako Chemicals).

**Ketone body level.** Plasma β-hydroxybutyrate concentrations were measured by using a β-Hydroxybutyrate Assay Kit (Abcam, ab83390).

**Fatty acid oxidation.** Primary hepatocytes isolated from $Crtc2^{f/f}$ mice or $Crtc2^{LKO}$ mice were seeded at $2 × 10^4$ cells per well on XF-24 cell culture plates (Seahorse Bioscience) in M199 medium (Sigma). After overnight incubation, cells were equilibrated with KHB buffer (111 mM NaCl, 4.7 mM KCl, 2 mM $MgSO_4$, 1.2 mM $Na_2HPO_4$, 2.5 mM glucose, 0.5 mM carnitine, pH 7.4) and incubated at 37 °C for 1 h without $CO_2$. For induction of fatty acid oxidation, palmitate–BSA complex was injected at a final concentration of 200 μM into the XF-24 cartridge (Seahorse Bioscience). Fatty acid oxidation capacity was represented as increased oxygen consumption rate in response to the palmitate–BSA complex.

**Adenovirus, AAV virus.** Adenovirus expressing constitutive active form of Crtc2 (Ad-Crtc2 S171A) and adenovirus expressing miR-34a (Ad-miR-34a) have been described previously[15, 23]. Recombinant adenovirus ($0.5 × 10^9$ pfu) was delivered by tail vein injection to the mice. For AAV-miR34a, miR-34a was first subcloned into the pAAV-IRES, and was co-transfected with pAAV-DJ vector and the pAAV-Helper vector into HEK 293T cells to generate recombinant adeno-associated virus expressing miR-34a as described in the manufacturer's protocol (Cell Biolabs). About $2 × 10^{10}$ vg (per mouse) of purified viruses was injected into the mice via tail vein injection. After 7 days, mice were fed a HFD for 7 weeks before being killed.

**Ribonucleoprotein immunoprecipition.** AGO2 immunoprecipitation experi-ments after Ad-miR-34a infection were conducted by using the liver tissues as described previously[39]. Briefly, the liver tissue was triturated in 10 ml of HBSS and transferred to a 10 cm dish. Tissue suspensions were irradiated three times at 400 mJ/cm² in the UV cross-linker. Cross-linked lysate was resuspended in 1×PXL (1× PBS, 0.1% SDS, 0.5% deoxycholate, 0.5% NP-40, w/o $Mg^{2+}$, $Ca^{2+}$, complete pro-tease inhibitor cocktail, RNasin (Promega)). The lysate was incubated with pre-made AGO2 antibody (2E12, Abnova, H00027161-M01)-Dynabead Protein A complex for 4 h at 4 °C. The antibody corresponding protein–RNA complex was pull down with a magnetic bar (Invitrogen).The total RNA in the complex is purified by miRNeasy Kit (Qiagen) and reverse transcribed using the miScript II RT Kit (Qiagen) to analyze the miRNA expression.

**Metabolic cage analysis.** Indirect calorimetry was assessed by using indirect calorimetric chamber (OxyletPro System, Harvard Apparatus; Panlab) according to the manufacturer's protocol[40]. Briefly, male 16-week-old, 8-week-HFD-fed $Crtc2^{f/f}$ mice, and $Crtc2^{LKO}$ mice were acclimated in the calorimetric chamber for 24 h prior to the measurement. $VO_2$ and $VCO_2$ were measured every 9 min for 4 h by an $O_2$ and $CO_2$ analyzer at a controlled flow rate of 600 ml/min. At each point of analysis, the installed software automatically calculated the RQ and EE. In addition, food intake and physical activity were also measured in the system. Data represent the single 24 h measurement.

**Body composition analysis.** Body composition was measured by using the NMR analyzer Bruker Minispec LF50 (Bruker Optics Inc.) at the Korea Mouse Pheno-typing Center (KMPC), Seoul National University, Seoul, Korea.

**Small RNA sequencing.** For small RNA sequencing, RNAs were extracted from liver tissues of HFD-fed $Crtc2^{f/f}$ mice and $Crtc2^{LKO}$ mice by using miRNeasy Mini Kit (Qiagen). Total RNA quality and quantity were verified spectrophotometrically and electrophoretically (Bioanalyzer 2100). Sequencing library construction with the Illumina TruSeq small RNA Sample Prep Kit was performed as described in the manufacturer's protocol. Libraries were sequenced on the Illumina MiSeq instru-ment. Sequence alignment and detection of known and novel microRNAs were performed by using miRDeep2 software algorithm. Prior to performing sequence alignment, the *Mus musculus* reference genome release mm10 was retrieved from UCSC genome browser and indexed using Bowtie (1.1.1), a bowtie for aligning sequencing reads to reference sequences. The pre-processed and clustered reads were aligned to *Mus musculus* reference genome. Those reads were then aligned to the *Mus musculus* matured and precursor miRNAs obtained from the miRBase v21. The miRDeep2 algorithm is based on the miRNA biogenesis model; it aligns reads to potential hairpin structures in a manner consistent with Dicer processing, and assigns scores that represent the probability that hairpins are true miRNA precursors. In addition to detecting known and novel miRNAs, miRDeep2 esti-mates their abundance. The experiment was performed by Macrogen, Seoul, Korea.

**Statistics.** Results are represented as either mean ± s.e.m.(for metabolites) or mean ± s.d. (for Q-PCR and luciferase assay). Comparison of different groups was carried out by using two-tailed unpaired Student's $t$ test for comparing two groups, or Tukey–Kramer multiple comparisons for comparing values among multiple groups as indicated in the figure legends.

**Data availability.** All the data and materials supporting this work are available upon reasonable request to the corresponding author. The data for the small RNA sequencing were deposited in the Sequence Read Archive (SRA), National Cancer Center for Biotechnology Information (accession number: SAMN07661813).

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

## Acknowledgements

We thank Professor Sung-Gil Chi (Korea University, Seoul, Korea) for pCMV-flag-p53 construct and Professor Jongsook Kim Kemper (University of Illinois, Champaign, IL, USA) for miR-34a adenovirus construct. We thank Professor Je Kyung Seong (Korea Mouse Phenotyping Center, Seoul National University, Seoul, Korea) for the help to utilize NMR analyzer Bruker Minispec LF50, and Professor Sung-Wook Chi (Korea University, Seoul, Korea) for the help to analyze AGO2-associated miR-34a. We also thank Sun Myung Park (Korea University, Seoul, Korea) for technical assistance and members of our laboratory for the critical reading of the manuscript. This work was supported by the National Research Foundation of Korea (grant nos.: NRF-2012M3A9B6055345, NRF-2015R1A5A1009024, NRF- 2015R1A2A1A01006687, and NRF-2017M3A9D5A01052447), funded by the Ministry of Science and ICT, Republic of Korea, and a grant from Korea University, Seoul, Republic of Korea.

## Author contributions

The project was conceived by S.-H.K. Experiments were designed and performed by H.-S. H., B.H.C., J.S.K., G.K. and S.-H.K., and were analyzed by H.-S.H. and S.-H.K. H.-S.H. and S.-H.K. wrote the paper.

## Additional information

**Competing interests:** The authors declare no competing financial interests.

