## [Peer Review File · Nature Communications]

Reviewers' comments:

Reviewer #1 (Remarks to the Author):

This study reports on the effect of liver specific deletion of CRTC2 on energy expenditure and reaches the conclusion that this deletion results in suppression of mirR-34 which induces SIRT1 and PPAR α in the liver, ultimately increasing FGF21 which leads to improved metabolic status of the animals. While the data are consistent with this formulation, the conclusion that the rise in FGF21 is instrumental in mediating the improved metabolic state seems unlikely. FGF21 appears only to be increased in mice fed a HFD and only in the fasted state. Furthermore, this increase, unlike multi-fold increases seen with amino acid deprivation (Laeger, et al) or consumption of a ketogenic diet (Badman, et al) or fasting (Inagaki and Badman) is very modest. At the mRNA level the difference between WT and KO mice is just over two-fold and at the serum level it is less than two-fold. Furthermore, Douris et al have shown that two fold increases of FGF21 in the periphery do not have metabolic effects. Consistent with FGF21 as an unlikely mediator is the fact that downstream targets of FGF21 are but minimally increased. For example, there is no increase in UCP1 in SC fat and the increase in DIO2 is very small. PRDM16 which does not respond to pharmacologic doses of FGF21 is quite induced in this model. So while the improved metabolic phenotype is intriguing, it is unlikely to be attributable to FGF21.

Given the beneficial metabolic effects of the deletion, was the response to glucagon assessed in the KO mice? This should be assessed.

Specific Comments:

Figure 1 data: Age and weight of animals at the time of study needs to be provided. Time on the HFD diet should be included. Was body fat assessed by MRI or DEXA scan. How was locomotor activity evaluated? Baseline activity seems very high for C57 black males. With regard to energy expenditure, what is being reported here? The data says kcal per day, how is this derived. Data should NOT be transformed 0.75 power (Tschoep, et al). This is used for interspecies comparison as opposed to with a species. What was the weight of the tested animals? If per animal energy expenditure is evaluated is Energy expenditure still increased? Repeated measures anova should be used to confirm significance for the body weight curves. Calories consumed per day should be reported. The text states that there was no change, however there is no indication if this is total consumption. Food intake should not be corrected for body weight.

Figure 2 data: The reported data for circulating FGF21 is presumably from fasted animals, this needs to be specified. Can the authors comment on the absence of any difference in PPAR α between wild type and KO animals since this pathway is invoked as participating in the phenotype? Increased BDH1 is shown. Are KO animals more ketotic? If so, might excreted ketones contribute to the phenotype? In BAT the difference in protein UCP1 expression may be due to including the third lane of WT animals. Does the difference persist if this lane is excluded?

Figure 3 data: Were miRNA levels measured in the fed or fasted state? State of the animals in terms of fed or fasted needs to be specified for many of the panels. In panel e, samples are taken in the fasted state; comparative data in fed state should be included. Is miR-34 regulated by fasting?

Figure 4 data: The suppression of FGF21 with transcriptional activation of miR-34A is noted. Again is this in fed or fasted animals? Does activation inhibit the rise in FGF21 seen with a more profound challenge, i.e. protein restriction or ketogenesis?

Reviewer #2 (Remarks to the Author):

In this paper, Han and coworkers demonstrate that hepatic CREB coactivator CRTC2 controls whole body energy metabolism in part through a miR-34a-FGF21 regulatory axis. Utilizing *in vivo* studies from newly generated CRTC2-LKO mice and viral transduction studies, as well as, molecular biological studies in cells, the authors present strong evidence that CRTC2 inhibits increased expression of miR-34a by directly binding to the miR-34a promoter and that induction of miR-34a results in inhibition of a SIRT1/PPAR α /FGF21 axis in the liver, which impacts on systemic lipid and glucose metabolism. The *in vivo* effect of chronic depletion of hepatic CRTC2 as shown in CRTC2-LKO mice on systemic energy metabolism is impressive. Overall, this is an interesting set of experiments with strong rationale and execution and convincing results. However, there are a few issues need to be addressed.

- In the discussion section (lines 169-175), the authors present a physiologically relevant hypothesis about the temporal regulation of glucose and lipid metabolism in response to time of fasting. Under normal physiological conditions, in response to short term fasting, CREB/CRTC2 is critical for gluconeogenesis, while CRTC2 induction of miR-34a delays fat oxidation by inhibiting SIRT1/PPAR α . Upon longer-term fasting, CREB/CRTC2 activity is attenuated, which results in decreased miR-34a and increased SIRT1/PPAR α levels and activities that increase fat oxidation.

To firmly support this hypothesis, the authors should examine the effect of fasting time (short and longer term fasting) on miR-34a levels in floxed and CRTC2-LKO mice.

- It has been shown that circulating FGF21 levels are elevated in obese patients and lab animals, suggesting FGF21 resistance in obesity (Zhang et al., *Diabetes*, 2008; Fisher et al., *Diabetes*, 2010). Further, in recent studies, miR-34a in obesity attenuates FGF21 signaling by directly targeting the FGF21 receptor complex including the obligate co-receptor β KL (Fu et al., *MCB*, 2014).

In the present manuscript, the authors show that hepatic depletion of CRTC2 results in increased FGF21 expression through a miR-34a-SIRT1-PPAR α axis. Have the authors should examine the effect on FGF21 sensitivity/signaling in addition to FGF21 expression? *In vivo* studies from CRTC2-LKO and floxed mice would be ideal, but experiments could be done in hepatocytes isolated from these mice treated with FGF21 to measure the levels of downstream kinases, such as p-ERK.

- In few places, the bibliography is not entirely correct. In line 131, a reference (Xu et al., *Nature Communications*, 2015) was cited for elevated miR-34a levels in HFD-fed mice, but these findings had been already reported in earlier studies (Lee et al., *JBC*, 2010; Fu et al., *PNAS* 2012; Choi et al., *Aging Cell* 2013). This needs to be corrected.

- In Supple Fig. 3-c, mRNA levels of Utk1 appear to be increased in CRTC2-LKO mice. Is this statistically significant? Also, ATG7 protein levels appeared to be decreased. Were these mice fasted (if so, for short term or longer term)?

- In Fig 4-d, it is written that occupancy of CREB/CRTC2 is detected but only CRTC2 occupancy is presented. Is occupancy of CREB changed in these mice?

- In Fig 4 e-g, what are the levels of exogenously expressed miR-34a? Are these are pathologically relevant levels (i.e., levels detected in obese mice)?

- In Fig. 4-g, the labeling for the Y-axis is missing.

Response to reviewers' comments

Reviewers' comments:

Reviewer #1 (Remarks to the Author):

This study reports on the effect of liver specific deletion of CRTC2 on energy expenditure and reaches the conclusion that this deletion results in suppression of mirR-34 which induces SIRT1 and PPAR α in the liver, ultimately increasing FGF21 which leads to improved metabolic status of the animals. While the data are consistent with this formulation, the conclusion that the rise in FGF21 is instrumental in mediating the improved metabolic state seems unlikely. FGF21 appears only to be increased in mice fed a HFD and only in the fasted state. Furthermore, this increase, unlike multi-fold increases seen with amino acid deprivation (Laeger, et al) or consumption of a ketogenic diet (Badman, et al) or fasting (Inagaki and Badman) is very modest. At the mRNA level the difference between WT and KO mice is just over two-fold and at the serum level it is less than two-fold. Furthermore, Douris et al have shown that two fold increases of FGF21 in the periphery do not have metabolic effects. Consistent with FGF21 as an unlikely mediator is the fact that downstream targets of FGF21 are but minimally increased. For example, there is no increase in UCP1 in SC fat and the increase in DIO2 is very small. PRDM16 which does not respond to pharmacologic doses of FGF21 is quite induced in this model. So while the improved metabolic phenotype is intriguing, it is unlikely to be attributable to FGF21.

→ We reasoned that the potential difference in the aforementioned studies and the current study is the duration of the FGF21 exposure. While other studies utilized more stringent conditions to elicit higher FGF21 expression, the duration of the diet was relatively short (between 2-4 weeks). On the other hand, in our model, depletion of CRTC2 led to the mild, but consistent increase in FGF21 throughout the life time. We managed the high fat diet for 7-11 weeks, thus the 2-fold increase in plasma FGF21 during the course of experiment may be enough to elicit the current metabolic phenotype in our study. While waiting for the decision on our manuscript, we generated double liver-specific knockout mice for CRTC2 and FGF21 (CRTC2/FGF21^{LKO} mice), to directly address the question whether increased FGF21 in CRTC2^{LKO} mice is the cause of the current phenotype in our revised manuscript. While the double deficiency of CRTC2 and FGF21 did not alter glucose metabolism compared with a single CRTC2 liver knockout (Supplementary Fig. 8a,b), we observed that hepatic double deficiency of CRTC2 and FGF21 led to the accumulation of lipid in the liver, BAT and WAT compared with the CRTC2^{LKO} mice (Fig. 4i and Supplementary Fig. 8e). In addition, UCP1 protein levels in BAT were also reduced in the double knockout mice (Fig. 4i and Supplementary Fig. 8f). These data suggest that indeed the metabolic phenotype that is related to the energy homeostasis or lipid metabolism is in large part via the increased FGF21 levels upon hepatic depletion of CRTC2 in mice.

Given the beneficial metabolic effects of the deletion, was the response to glucagon assessed in the KO mice? This should be assessed.

→ We indeed performed the glucagon tolerance test on CRTC2^{ff} mice and CRTC2^{LKO} mice but did not include in the original manuscript. We added the data in the revised manuscript, which showed that CRTC2^{LKO} mice exhibited reduced plasma glucose levels in response to glucagon (Supplementary Fig. 3a)

Specific Comments:

Figure 1 data: Age and weight of animals at the time of study needs to be provided. Time on the HFD diet should be included.

→ We provided the information regarding the age of the animals, as well as the duration of the HFD on each experiment in our revised figure legends. Please refer to the figure 1f for the weight of animals.

Was body fat assessed by MRI or DEXA scan.

→ We measured the body composition of $CRTC2^{ff}$ mice and $CRTC2^{LKO}$ mice by using NMR analyzer Bruker Minispec LF50 (Bruker Optics Inc.) (Supplementary Fig. 3e). Please note that due to the limitation of the time for the revision (three months were allowed by the editor), we used older mice that we maintained under the high fat diet (23-week) for the test in the revised manuscript.

How was locomotor activity evaluated? Baseline activity seems very high for C57 black males.

→ We would apologize for the confusion. In the previous version, we added both the locomotor activity and the rearing activity. In our revised manuscript, only the locomotor activity was shown (Fig. 1g). The measurement was detected as beam breaks for the indicated time period as shown in the manual of OxyletPro™ System.

With regard to energy expenditure, what is being reported here? The data says kcal per day, how is this derived.

→ The base setting for the indirect calorimetric equipment varies among different manufactures and models. In our model (OxyletPro™ System), it was shown as kcal/day as opposed to kcal/h or g/h etc. I suspect it is just a different way to express the data. We could easily switch it to kcal/h, although it requires a re-calculation and will not add any more information. Nonetheless, heeding the reviewer's advice, we re-express our data as kcal/d in our revised manuscript (Fig. 1g).

Data should NOT be transformed 0.75 power (Tschoep, et al). This is used for interspecies comparison as opposed to with a species. What was the weight of the tested animals? If per animal energy expenditure is evaluated is Energy expenditure still increased?

→ I would appreciate the reviewer's comment and suggestions regarding the matter. We re-express our data on energy expenditure per animals, without regard to the individual weights (kcal/d). We observed that the difference of EE between WT and $CRTC2^{LKO}$ mice are still significant. We replaced the new analysis in our revised manuscript (Fig. 1g).

Repeated measures anova should be used to confirm significance for the body weight curves.

→ We performed the 2-way ANOVA test for entire HFD-fed period and found that body weights of $CRTC2^{ff}$ mice and $CRTC2^{LKO}$ mice are significantly different ($p < 0.0001$) (Fig. 1f)

Calories consumed per day should be reported. The text states that there was no change, however there is no indication if this is total consumption. Food intake should not be corrected for body weight.

→ We re-calculate the food intake data by not correcting them with the body weight in

our revised manuscript as suggested by the reviewer, and presented as the food intake per day (g/d), which is not significantly different between the two groups (Supplementary Fig. 4d).

Figure 2 data: The reported data for circulating FGF21 is presumably from fasted animals, this needs to be specified.

→ We would apologize for not specifying the condition. The data were indeed from the fasted animals, and were now shown in the figure legend.

Can the authors comment on the absence of any difference in PPARα between wild type and KO animals since this pathway is invoked as participating in the phenotype?

→ Since we proposed that the depletion of CRTC2 in the liver reduced miR-34a that affects PPAR α protein, we would expect to observe more significant changes in protein levels, but not mRNA levels of PPAR α.

Increased BDH1 is shown. Are KO animals more ketotic? If so, might excreted ketones contribute to the phenotype?

→ We measured the plasma ketone body (beta-hydroxybutyrate) and found that it is higher in the CRTC2^{LKO} mice (Supplementary Fig. 4e). It is possible that a slight increase in ketone body in the plasma affects the metabolic phenotype, but it could be also correlated with the increased FGF21 levels, since we observed the reduced plasma ketone body in the CRTC2/FGF21^{LKO} mice (Fig. 4e).

In BAT the difference in protein UCP1 expression may be due to including the third lane of WT animals. Does the difference persist if this lane is excluded?

→ We re-calculated the protein bands without the third lane, and we were still able to observe the difference (In old data, it was 1 ± 0.2458 (CRTC2^{ff} mice) vs 1.4535 ± 0.1308 (CRTC2^{LKO} mice) (P=0.0173). If we drop the third lane of CRTC2^{ff} mice, the difference will be smaller, but the difference between two samples will be more significant due to the smaller variations in values for CRTC2^{ff} mice (1 ± 0.0815 (CRTC2^{ff} mice) vs. 1.3011 ± 0.1171 (CRTC2^{LKO} mice) (P=0.01290)). In addition, we observed more compelling data on UCP1 protein levels in BAT (please compare the UCP1 protein levels among WT, CRTC2^{LKO} mice, and CRTC2/FGF21^{LKO} mice) (Fig. 4i and Supplementary Fig. 8f)

Figure 3 data: Were miRNA levels measured in the fed or fasted state? State of the animals in terms of fed or fasted needs to be specified for many of the panels.

→ We would apologize for not specifying the condition. The data were indeed from the fasted animals, and the condition was now shown in the figure legend.

In panel e, samples are taken in the fasted state; comparative data in fed state should be included.

→ We provided the mRNA data both under fasting and feeding in the new experiment shown in Supplementary Fig. 6b,c.

Is miR-34 regulated by fasting?

→ When we compare liver samples from the overnight fasting and feeding conditions

under 9 week-HFD, we did not observe specific differences in miR-34a expression between the fasting and ad libitum conditions (Reviewer Fig 1). We presumed that under diet-induced obesity and insulin resistant conditions, consistent activation of CREB/CRTC2 (both under feeding and fasting)¹ could promote higher expression of miR-34a in the liver, which contributes to the increased expression of hepatic miR-34a upon HFD feeding (as shown Supplementary Fig. 5f and other recent reports^{2, 3, 4, 5}). Thus, we revised the final paragraph and omit the potential involvement of CREB/CRTC2 in the control of PPAR alpha signaling via miR-34a pathway under the normal physiological conditions. We also revised the sentences regarding the transcriptional control of miR-34a accordingly in the revised manuscript.

Figure 4 data: The suppression of FGF21 with transcriptional activation of miR-34A is noted. Again is this in fed or fasted animals?

→ We would apologize for not specifying the condition. The data were indeed from the fasted animals, and the condition was shown in the revised manuscript.

Does activation inhibit the rise in FGF21 seen with a more profound challenge, i.e. protein restriction or ketogenesis?

→ We fed animals with ketogenic diet for one week and then infected with adenovirus for an additional week to see whether the effect of CREB/CRTC2 on FGF21 was more pronounced. We observed more pronounced reduction of plasma FGF21 as well as FGF21 mRNA levels by S171A in the liver (Reviewer Fig. 2).

Reviewer #2 (Remarks to the Author):

In this paper, Han and coworkers demonstrate that hepatic CREB coactivator CRTC2 controls whole body energy metabolism in part through a miR-34a-FGF21 regulatory axis. Utilizing in vivo studies from newly generated CRTC2-LKO mice and viral transduction studies, as well as, molecular biological studies in cells, the authors present strong evidence that CRTC2 inhibits increased expression of miR-34a by directly binding to the miR-34a promoter and that induction of miR-34a results in inhibition of a SIRT1/PPAR α /FGF21 axis in the liver, which impacts on systemic lipid and glucose metabolism. The in vivo effect of chronic depletion of hepatic CRTC2 as shown in CRTC2-LKO mice on systemic energy metabolism is impressive. Overall, this is an interesting set of experiments with strong rationale and execution and convincing results. However, there are a few issues need to be addressed.

- In the discussion section (lines 169-175), the authors present a physiologically relevant hypothesis about the temporal regulation of glucose and lipid metabolism in response to time of fasting. Under normal physiological conditions, in response to short term fasting, CREB/CRTC2 is critical for gluconeogenesis, while CRTC2 induction of miR-34a delays fat oxidation by inhibiting SIRT1/PPAR α . Upon longer-term fasting, CREB/CRTC2 activity is attenuated, which results in decreased miR-34a and increased SIRT1/PPAR α levels and activities that increase fat oxidation.

To firmly support this hypothesis, the authors should examine the effect of fasting time (short and longer term fasting) on miR-34a levels in floxed and CRTC2-LKO mice.

→ Due to the time limitation for the revision (3 months), we were unable to test the effects of fasting time in HFD-fed mice. However, when we just compare overnight fasting and feeding conditions under 9 week-HFD, we did not observe specific differences in miR-34a expression between the conditions (Reviewer Fig 1). Besides,

when we utilized the normal chow-diet mice for the study, we were not able to observe the consistent changes in hepatic miR-34a levels by fasting or even the depletion of liver-specific CRTC2 in mice (data not shown). We suspect that cAMP-dependent transcriptional input *per se* might be necessary but not sufficient to promote increased expression of miR-34a, at least under the normal diet conditions. We presumed that under diet-induced obesity and insulin resistant conditions, consistent activation of CREB/CRTC2 (both under feeding and fasting)¹ could promote higher expression of miR-34a in the liver, which contributes to the increased expression of hepatic miR-34a upon HFD feeding. We could also mimic such conditions by using adenovirus expressing constitutively active CRTC2 (CRTC2 S171A) (Fig. 3d,e). Thus, we revised the final paragraph and omit the potential involvement of CREB/CRTC2 in the control of PPAR alpha signaling via miR-34a pathway under the normal physiological conditions. We also revised the sentences regarding the transcriptional control of miR-34a accordingly in the revised manuscript.

- It has been shown that circulating FGF21 levels are elevated in obese patients and lab animals, suggesting FGF21 resistance in obesity (Zhang et al., Diabetes, 2008; Fisher et al., Diabetes, 2010). Further, in recent studies, miR-34a in obesity attenuates FGF21 signaling by directly targeting the FGF21 receptor complex including the obligate co-receptor β KL (Fu et al., MCB, 2014).

In the present manuscript, the authors show that hepatic depletion of CRTC2 results in increased FGF21 expression through a miR-34a-SIRT1-PPAR α axis. Have the authors should examine the effect on FGF21 sensitivity/signaling in addition to FGF21 expression? In vivo studies from CRTC2-LKO and floxed mice would be ideal, but experiments could be done in hepatocytes isolated from these mice treated with FGF21 to measure the levels of downstream kinases, such as p-ERK.

➔ We measured p-ERK levels both in mouse liver lysates that were used in Fig. 1 (CRTC2^{ff} mice and CRTC2^{LKO} mice). We observed elevations in p-ERK levels in the liver of CRTC2^{LKO} mice compared with the control, showing the increased FGF21 effects in the liver (Reviewer Fig. 3).

- In few places, the bibliography is not entirely correct. In line 131, a reference (Xu et al., Nature Communications, 2015) was cited for elevated miR-34a levels in HFD-fed mice, but these findings had been already reported in earlier studies (Lee et al., JBC, 2010; Fu et al., PNAS 2012; Choi et al., Aging Cell 2013). This needs to be corrected.

➔ We would apologize for not correctly citing the appropriate papers. We corrected our mistake in our revised manuscript.

- In Supple Fig. 3-c, mRNA levels of Ulk1 appear to be increased in CRTC2-LKO mice. Is this statistically significant? Also, ATG7 protein levels appeared to be decreased. Were these mice fasted (if so, for short term or longer term)?

➔ These mice were fasted for 16-hr before being sacrificed as stated in our revised manuscript. Indeed, ULK1 mRNA levels were significantly higher in CRTC2^{LKO} mouse liver compared with the control. While ATG7 protein levels were also significantly reduced, protein levels of ATG16L1 were increased upon liver-specific depletion of

CRTC2. In addition, we did not observe any significant difference in p62 or LC3-II levels between CRTC2^{LKO} mice and the control mice. These data could suggest that changes in the autophagy might not be directly responsible for the phenotype shown by liver-specific depletion of CRTC2. We revised the sentences related to the description of data (now in supplementary Fig. 4 a,b) in the revised manuscript.

• In Fig 4-d, it is written that occupancy of CREB/CRTC2 is detected but only CRTC2 occupancy is presented. Is occupancy of CREB changed in these mice?

➔ In our revised manuscript, we provided the CREB occupancy over miR-34a promoter (Fig. 3g)

• In Fig 4 e-g, what are the levels of exogenously expressed miR-34a? Are these are pathologically relevant levels (i.e., levels detected in obese mice)?

➔ We measured the hepatic miR-34a levels upon AAV injection, and found that they were 4 fold higher compared with the WT mice (Fig. 4c). It appeared that the level of the overexpression might be within the range that you would expect to observe in the pathological condition. (In Supplementary Fig. 5f, we observed about 2.5-fold induction of miR-34a levels under HFD compared with the control. Others observed varying degree of increase in miR-34a expression by HFD-feeding (4.5-fold by Fu et. al.³, 2-fold by Xu et. al.⁵, 3-fold by Lee et. al.⁴, and 7-fold by Choi et. al.².)

• In Fig. 4-g, the labeling for the Y-axis is missing.

➔ We would apologize for our mistake. We provided the correct labeling for the Y-axis (now in Fig. 4d).

References

1. Dentin R, Liu Y, Koo SH, Hedrick S, Vargas T, Heredia J, *et al.* Insulin modulates gluconeogenesis by inhibition of the coactivator TORC2. *Nature* 2007, **449**(7160): 366-369.
2. Choi SE, Fu T, Seok S, Kim DH, Yu E, Lee KW, *et al.* Elevated microRNA-34a in obesity reduces NAD⁺ levels and SIRT1 activity by directly targeting NAMPT. *Aging cell* 2013, **12**(6): 1062-1072.
3. Fu T, Choi SE, Kim DH, Seok S, Suino-Powell KM, Xu HE, *et al.* Aberrantly elevated microRNA-34a in obesity attenuates hepatic responses to FGF19 by targeting a membrane coreceptor beta-Klotho. *Proceedings of the National Academy of Sciences of the United States of America* 2012, **109**(40): 16137-16142.
4. Lee J, Padhye A, Sharma A, Song G, Miao J, Mo YY, *et al.* A pathway involving farnesoid X receptor and small heterodimer partner positively regulates hepatic sirtuin 1 levels via microRNA-34a inhibition. *The Journal of biological chemistry* 2010, **285**(17): 12604-12611.
5. Xu Y, Zalzal M, Xu J, Li Y, Yin L, Zhang Y. A metabolic stress-inducible miR-34a-HNF4alpha pathway regulates lipid and lipoprotein metabolism. *Nature communications* 2015, **6**: 7466.

Reviewer Figures

Reviewer Figure 1. Effects of hepatic CRTC2 knockout on mature miR-34a expression in the livers of ad libitum or 16 h-fasted, 9 week-HFD-fed mice (Q-PCR, n=4 mice per group). Data represent mean \pm SD (**; $P < 0.01$, t-test).

Reviewer Figure 2. Effects of expression of constitutively active form of CRTC2 (S171A) on hepatic genes. 8 week-old C57/BL6 mice that were under the ketogenic diet for 1 week were injected with adenovirus for 5 days before being sacrificed after 24 h fasting. Plasma FGF21 levels (left) and hepatic FGF21 mRNA levels (right) were shown. Data in the left represent mean \pm SEM (*; $P < 0.05$, t-test), and data in the right represent mean \pm SD (**; $P < 0.01$, t-test).

Reviewer Figure 3. Effects of hepatic CRTC2 knockout on ERK and phosphor-ERK levels in the livers of 16 h-fasted, 9 week-HFD-fed mice.

Reviewers' comments:

Reviewer #1 (Remarks to the Author):

This is a revised manuscript examining the effect of liver specific deletion of the transcriptional co-activator CRTC2 metabolism.

The manuscript is convincing in demonstrating that liver specific deletion of CRTC2, in the setting consumption of a high fat diet leads to lower weight and improved metabolic status. In particular glucoses are lower, there is enhanced insulin sensitivity as assessed by glucose lowering after insulin administration and a lowering of liver triglycerides. These effects are associated with an activation of PPAR α resulting from decreased expression of the microRNA miR-34A. In addition increased fatty acid oxidation is shown, but only in the fasted state. There is a very small increase in overall energy expenditure which appears to be about 3.5 %.

What is not convincing is that the small rise in FGF21 noted in mice consuming a high fat diet mediates the positive metabolic changes. Adding the liver specific FGF21-KO does not add weight to the conclusion, in part because data seems to be shown selectively.

While the authors respond that other studies of diet on FGF21 expression were short, and that the chronic effects of activation over the entire duration over the 10 week time span of the high fat diet explains the phenotype, this response isn't to the point. The increase in FGF21 expression is still small, and further confounded by the fact that obesity is an FGF21 resistant state.

Furthermore, small effects can be seen with very short term infusion of relatively low doses when for very short periods (days) when FGF21 is administered ICV (see Douris et al, Endocrinology 2015). One of the original criticisms of this reviewer is that the increase in FGF21 is only reported in the fasted state -- after 16 hours of food deprivation and that the increase is relatively small -- i.e. 35%. Mice don't normally fast, they eat mainly during the dark cycle but also snack during the day. The data after a 16 hour fast likely reflect increased activity of PPAR α downstream genes in the context of calorie deprivation, but there is no evidence that during the course of a typical 24 hour light/dark feeding cycle FGF21 is ever increased.

Thus, one of the problems in interpreting the data, whether short term or long term is selective reporting. Rather than telling the whole story, what happens in fed and fasted animals under all sets of circumstances the paper only reports on FASTED levels in mice fed DIO.

Data on gene expression shows no difference in the fed state. The absence of a difference is a bit perplexing as the KO animals are leaner and have lower degrees of fatty liver and lower FGF21 would be expected. Serum samples must be available so fed FGF21 levels should be provided. As obesity leads to FGF21 resistance, leaner mice should have increased FGF21 sensitivity, hence the increase in circulating FGF21 is counterintuitive.

The evidence of increased scWAT and BAT activity is intriguing but inconsistent. PRDM16 is not a marker of browning in scWAT. At the protein level, the increases in UCP1 in scWAT is small and even smaller in BAT. How is UCP1 expression corrected? Lean mice have more protein to fat ratio and when looking relative protein expression needs to be corrected to the total protein content of the tissue. How does relative UCP1 protein expression compare in HFD fed animals compared to chow fed animals. FGF21 is expressed in both BAT and scWAT. Has FGF21 expression been evaluated in these tissues?

Data on the combined liver specific CRTC2KO/FGF21KO show a loss of the effect of CRTC2KO deletion. However, FGF21 intrinsically regulates fatty acid oxidation in the liver. Do liver specific FGF-21KO animals fed a HFD show increased liver triglyceride independent of CRTC2 deletion. This would be a predicted consequence of hepatic FGF21 deletion. Thus the increase in liver in liver fat in the combined knockout may be entirely independent of CRTC2.

In the regard to double KO animals, the absence of UCP1 expression in subcutaneous BAT is entirely perplexing.

Overall the FGF21-liver specific KO is not sufficiently characterized, in and of itself to be able to draw any conclusions about the effect of actions outside the liver.

Additional comments: Why is energy expenditure presented as kcal/day in figure 1g and plotted against an hourly axis? There are two problems with this. EE is a calculated number that frequently includes weight in the calculation. Leaner animals may well have increased energy

expenditure based on a smaller denominator. Secondly what calculation leads to being able to express EE as Kcal/Day if the X axes is plotted in hours? In the supplementary diet VO₂ is used instead. Why not here?

In supplementary Figure 3 animals show a decreased response to glucagon. What's the liver glycogen? The impaired response is to be expected if liver glycogen is low. Some of the changes, including decreased hepatic fatty acid synthesis (eg Supplementary figure 3D) could well be due to improved insulin sensitivity.

Reviewer #2 (Remarks to the Author):

This revised manuscript has been substantially improved and all issues raised have been appropriately addressed.

Response to reviewers' comments

Reviewers' comments:

Reviewer #1 (Remarks to the Author): This is a revised manuscript examining the effect of liver specific deletion of the transcriptional co-activator CRTC2 metabolism. The manuscript is convincing in demonstrating that liver specific deletion of CRTC2, in the setting consumption of a high fat diet leads to lower weight and improved metabolic status. In particular glucoses are lower, there is enhanced insulin sensitivity as assessed by glucose lowering after insulin administration and a lowering of liver triglycerides. These effects are associated with an activation of PPAR α resulting from decreased expression of the microRNA miR-34A. In addition increased fatty acid oxidation is shown, but only in the fasted state. There is a very small increase in overall energy expenditure which appears to be about 3.5 %.

What is not convincing is that the small rise in FGF21 noted in mice consuming a high fat diet mediates the positive metabolic changes. Adding the liver specific FGF21-KO does not add weight to the conclusion, in part because data seems to be shown selectively.

While the authors respond that other studies of diet on FGF21 expression were short, and that the chronic effects of activation over the entire duration over the 10 week time span of the high fat diet explains the phenotype, this response isn't to the point. The increase in FGF21 expression is still small, and further confounded by the fact that obesity is an FGF21 resistant state. Furthermore, small effects can be seen with very short term infusion of relatively low doses when for very short periods (days) when FGF21 is administered ICV (see Douris et al, Endocrinology 2015). One of the original criticisms of this reviewer is that the increase in FGF21 is only reported in the fasted state -- after 16 hours of food deprivation and that the increase is relatively small -- i.e. 35%. Mice don't normally fast, they eat mainly during the dark cycle but also snack during the day. The data after a 16 hour fast likely reflect increased activity of PPAR α downstream genes in the context of calorie deprivation, but there is no evidence that during the course of a typical 24 hour light/dark feeding cycle FGF21 is ever increased. Thus, one of the problems in interpreting the data, whether short term or long term is selective reporting. Rather than telling the whole story, what happens in fed and fasted animals under all sets of circumstances the paper only reports on FASTED levels in mice fed DIO.

→ We indeed performed three independent measurements of FGF21 levels during fasting and feeding, and presented a set of representative data. We did not observe specific changes in feeding FGF21 levels between the two genotypes (Reviewer Figure a). Thus, we believe that the FGF21 levels are increased in CRTC2 liver-specific knockout mice in a fasting status-specific manner. Being regarded as an endocrine hormone, this type of regulation of FGF21 levels are not unprecedented. Indeed, most endocrine hormones are regulated in response to the nutrient status. Examples include insulin (during feeding), glucagon (during fasting), and many incretins such as GLP-1 and GIP-1 (during feeding), which mainly function during the specific nutrient status and countered by counter-regulatory hormones upon changes in nutrient state. While the

secretion of such hormones are regulated at the level of exocytosis, there are examples of endocrine hormones such as GLP-1 that are also regulated at the level of transcription. In addition, counter-regulatory hormones exist to limit the action of certain hormones for a prolonged period of time. CRTC2 liver-specific knockout mice showed increased insulin sensitivity, thus we suspected that insulin could counteract to reduce FGF21 levels during feeding, by reducing transcriptional activation of hepatic FGF21 via inhibition of PPAR alpha activity.

Data on gene expression shows no difference in the fed state. The absence of a difference is a bit perplexing as the KO animals are leaner and have lower degrees of fatty liver and lower FGF21 would be expected. Serum samples must be available so fed FGF21 levels should be provided. As obesity leads to FGF21 resistance, leaner mice should have increased FGF21 sensitivity, hence the increase in circulating FGF21 is counterintuitive.

➔ In our view, the increased FGF21 levels in CRTC2 liver-specific mice are due to the increased expression of hepatic FGF21 via PPAR alpha-dependent transcriptional process. Hence, we suspected that we observed higher FGF21 levels (and hepatic expression) during fasting state of the HFD-fed CRTC2 liver-specific knockout mice in spite of the improved adiposity. We also measured p-ERK levels in livers of WT and CRTC2 liver-specific knockout mice (Reviewer Figure b). We found that p-ERK levels were higher in livers of KO mice compared with the control, showing an increased FGF21 action in the liver either due to the increased FGF21 sensitivity, the increased FGF21 levels, or both.

The evidence of increased scWAT and BAT activity is intriguing but inconsistent. PRDM16 is not a marker of browning in scWAT.

➔ Heeding the advice from the reviewer, we removed Q-PCR data for BAT and scWAT in our revised manuscript to avoid confusion.

At the protein level, the increases in UCP1 in scWAT is small and even smaller in BAT. How is UCP1 expression corrected? Lean mice have more protein to fat ratio and when looking relative protein expression needs to be corrected to the total protein content of the tissue. How does relative UCP1 protein expression compare in HFD fed animals compared to chow fed animals.

➔ UCP1 protein expression was corrected with the loading control, HSP90. We collected tissues, and prepared tissue extracts, measured the protein concentration, and loaded the same amount of the protein, following the standard protocol. In our knowledge, one cannot accurately and quantitatively measure the total protein content of the tissue, unless he or she uses up all the tissues for the preparation of proteins only, which is hardly practical.

It may well be possible that lean mice have more protein to fat ratio, but we specifically extracted protein, not fat, for our preparation. To address the issue, we performed western blot analysis by using BAT from HFD-fed mice and CD-fed mice (both in C57/BL6 wild type background). As shown in the Reviewer Figure c, we saw rather

elevations in UCP1 protein levels in BAT from HFD-fed mice compared with the control. In compliance with our data, we found two recent publications where the authors also observed the increased BAT UCP1 protein levels by HFD^{1, 2}. Thus, we suspect that normalizing the UCP1 protein with the loading control is adequate. As for detecting UCP1, we used 60 µg of protein from scWAT extracts, while we used 10 µg of protein from BAT extracts in accordance with the previous reports regarding the abundance of UCP1 in each tissue.

FGF21 is expressed in both BAT and scWAT. Has FGF21 expression been evaluated in these tissues?

➔ We performed Q-PCR analysis from BAT and scWAT from WT and KO mice. We found no significant differences in FGF21 mRNA levels between two genotypes (Reviewer Figure d).

Data on the combined liver specific CRTC2KO/FGF21KO show a loss of the effect of CRTC2KO deletion. However, FGF21 intrinsically regulates fatty acid oxidation in the liver. Do liver specific FGF-21KO animals fed a HFD show increased liver triglyceride independent of CRTC2 deletion. This would be a predicted consequence of hepatic FGF21 deletion. Thus the increase in liver in liver fat in the combined knockout may be entirely independent of CRTC2.

➔ To address this issue, we freshly bred mice to prepare WT (double flox/flox mice), CRTC2 liver-specific knockout mice, FGF21 liver-specific knockout mice, and CRTC2/FGF21 double liver-specific knockout mice and fed them high fat diet for 10 weeks (Supplementary Fig. 8). As shown before, CRTC2 liver-specific knockout mice showed reduced adiposity as evidenced by reduced fat cell size and hepatic lipid accumulation compared with the wild type control. The beneficial effect of CRTC2 depletion was reversed by concomitant knockout of FGF21 in the liver. However, we did not observe any specific phenotype of FGF21 single liver-specific knockout mice compared with the control. We suspected that increased FGF21 in CRTC2 LKO mice was indeed beneficial in combating HFD-induced obesity, and the depletion of FGF21 in this background simply reversed this phenotype. On the other hand, simple depletion of hepatic FGF21 *per se* might not be enough to promote dysregulation in lipid homeostasis, at least in our hands.

In the regard to double KO animals, the absence of UCP1 expression in subcutaneous BAT is entirely perplexing.

➔ In that specific blot, we observed dramatically reduced UCP1 levels in BAT of double knockout mice, although UCP1 IHC showed some expression in that tissue. We removed previous data and replaced them with new results from experiments using WT (double flox/flox mice), CRTC2 liver-specific knockout mice, FGF21 liver-specific knockout mice, and CRTC2/FGF21 double liver-specific knockout mice in our revised manuscript (Supplementary Fig. 8).

Overall the FGF21-liver specific KO is not sufficiently characterized, in and of itself to be able to draw any conclusions about the effect of actions outside the liver.

➔ We agree that this set of experiments may not unequivocally show that the effect of CRTC2 liver-specific knockout mice on adiposity and energy metabolism is due to the altered miR-34a-FGF21 axis. To address this issue, we performed the add-back experiment with AAV-FGF21 in CRTC2 LKO/miR-34a mice to show that the effect of miR-34a on adiposity and hepatic steatosis in mice could be rescued by the co-expression of FGF21 (Fig 4 and Supplementary Fig 9).

Additional comments: Why is energy expenditure presented as kcal/day in figure 1g and plotted against an hourly axis? There are two problems with this.

EE is a calculated number that frequently includes weight in the calculation. Leaner animals may well have increased energy expenditure based on a smaller denominator.

➔ In our original submission, we calculated EE as kcal/d/kg^{0.75}, to normalize the overall changes by body weight. In the revised manuscript, to exclude some of the concerns regarding the differences in body weight between the two genotypes, we omitted the normalization by body weight, showing a smaller but consistently higher levels of EE for CRTC2 liver-specific knockout mice compared with the control. Thus, we did not include weight in the calculation by accepting the reviewer's previous suggestion (Fig. 1g).

Secondly what calculation leads to being able to express EE as Kcal/Day if the X axes is plotted in hours? In the supplementary diet VO₂ is used instead. Why not here?

➔ As we have previously stated, in our model (OxyletPro™ System), it was shown as kcal/day as opposed to kcal/h or g/h etc. Heeding the reviewer's advice, we re-calculated the data and presented as kcal/h in the revised manuscript (Fig. 1g).

In supplementary Figure 3 animals show a decreased response to glucagon. What's the liver glycogen? The impaired response is to be expected if liver glycogen is low.

➔ We measured liver glycogen levels between the two genotypes, found no significant differences between them under feeding conditions, but observed a reduced hepatic glycogen levels in knockout mice compared with the control under fasting conditions (Supplementary Fig. 3b). Since we fasted mice for 6 h for the glucagon challenge test (Supplementary Fig. 3a), we would agree that we saw reduced glucose levels in CRTC2 liver-specific knockout mice in response to glucagon due to the lower liver glycogen levels.

Some of the changes, including decreased hepatic fatty acid synthesis (eg Supplementary figure 3D) could well be due to improved insulin sensitivity.

→ It is possible that the reduced fatty acid synthesis could be due to the improved insulin sensitivity. We added such comments in the revised manuscript.

Reviewer #2 (Remarks to the Author):

This revised manuscript has been substantially improved and all issues raised have been appropriately addressed.

→I would appreciate the kind consideration regarding our manuscript.

Reviewer Figure

a

b

c

d

Reviewer Figure Legend

a. Effects of chronic depletion of hepatic CRTC2 on plasma FGF21 levels in 16 h-fasted mice or ad libitum-fed mice under HFD for 10 weeks (n=7~11 mice per group). **b.** Effects of hepatic CRTC2 knockout on protein levels of ERK and p-ERK in the livers of 16 h-fasted, 9 week-HFD-fed mice. **c.** Effects of HFD on UCP1 protein expression in BAT. Mice were fed on normal chow diet (NCD) or high fat diet (HFD) for 10 weeks. **d.** Effects of chronic depletion of hepatic CRTC2 in mice under HFD for 9 weeks on FGF21 expression in scWAT and BAT (Q-PCR, n=5~7 mice per group).

References

1. **Fromme T, Klingenspor M. Uncoupling protein 1 expression and high-fat diets. *American journal of physiology Regulatory, integrative and comparative physiology* 2011, 300(1): R1-8.**
2. **Garcia-Ruiz E, Reynes B, Diaz-Rua R, Ceresi E, Oliver P, Palou A. The intake of high-fat diets induces the acquisition of brown adipocyte gene expression features in white adipose tissue. *Int J Obesity* 2015, 39(11): 1619-1629.**

REVIEWERS' COMMENTS:

Reviewer #1 (Remarks to the Author):

Many of my questions have been addressed.

The following questions remain:

1 - it is not clear how statistics were done for each set of reported figures. The figure legends are not specific enough. These need to be clarified better. For example, figures 1, panels a. It looks like T-tests were done at each individual time point. Appropriate to these tests is area under the curve. For Figure 1, panel G, does this represent single 24 hour monitoring period? Were mice acclimated in monitoring cages, for how long? What is the temperature of the facility where this monitoring is performed

2 - The figure provided to this reviewer must be added to the paper. It is misleading to suggest that FGF21 is always elevated relative to control mice when in fact it is only elevated when mice are fasted.

3 - The authors are unclear as to why the liver in their knockout mice have decreased lipid. Liver fat is regulated largely by the level of fatty acid synthesis and lipogenesis or by fatty acid oxidation. They seem to be unaware -- or fail to refer to studies showing that FGF21 has direct effects to increase fatty acid oxidation in whole animals. They don't refer to work by either Fisher et al or Tanaka et al to this regard and that the downstream consequences of increased fatty acid oxidation are decreased liver fat, decreased hepatic inflammation and potentially improved insulin sensitivity as a result. This needs to be added as a potential mechanism and should be added to the summary in Figure 4D.

The 2015 review cited minimizes effects of FGF21 on the liver which have since been acknowledged and reviewed by others. In listing transcription factors that regulate FGF21, ChREBP needs to be added. This has been reported by several, most recently by Fisher in molecular metabolism.

Reviewer #2 (Remarks to the Author):

In this revised study, with additional new data, the authors convincingly show that the prolonged CRT2 activity, under insulin resistance, not only activates hepatic glucose production, but also promotes lipid accumulation by activation of a miR-34a-SIRT1/PPAR α -FGF21 axis. Overall, this is an excellent study that will advance our understanding of the metabolic function of CRT2.

Response to reviewers' comments

REVIEWERS' COMMENTS:

Reviewer #1 (Remarks to the Author):

Many of my questions have been addressed.

→ I sincerely appreciate the reviewer's comments regarding our manuscript. I hope that our revised manuscript would successfully address the remaining concerns regarding our study.

The following questions remain:

1 - it is not clear how statistics were done for each set of reported figures. The figure legends are not specific enough. These need to be clarified better. For example, figures 1, panels a. It looks like T-tests were done at each individual time point. Appropriate to these tests is area under the curve.

→ We revised our figure legends to specifically show which statistical analysis was performed in each experiment. We also added the area under the curve data for Fig. 1a in our revised manuscript.

For Figure 1, panel G, does this represent single 24 hour monitoring period? Were mice acclimated in monitoring cages, for how long? What is the temperature of the facility where this monitoring is performed

→ To acclimate in the new environment, mice were housed in the calorimetric chamber (housed at 22°C) for 24 hours prior to the measurement. Data represent the single 24 hour measurement. These information regarding the indirect calorimetry analysis is now shown in the methods section of our revised manuscript.

2 - The figure provided to this reviewer must be added to the paper. It is misleading to suggest that FGF21 is always elevated relative to control mice when in fact it is only elevated when mice are fasted.

→ We replaced the Fig. 2f to show that the elevation of FGF21 by hepatic CRT2 depletion is only observed during fasting in our revised manuscript.

3 - The authors are unclear as to why the liver in their knockout mice have decreased lipid. Liver fat is regulated largely by the level of fatty acid synthesis and lipogenesis or by fatty acid oxidation. They seem to be unaware -- or fail to refer to studies showing that FGF21 has direct effects to increase fatty acid oxidation in whole animals. They don't refer to work by either Fisher et al or Tanaka et al to this regard and that the downstream consequences of increased fatty acid oxidation are

decreased liver fat, decreased hepatic inflammation and potentially improved insulin sensitivity as a result. This needs to be added as a potential mechanism and should be added to the summary in Figure 4D.

→ We would suspect that increased SIRT1/PPAR α program could enhance fatty acid oxidation in the liver (as shown in isolated primary hepatocytes), which reduces fatty liver symptoms in HFD-fed CRTC2 liver-specific knockout mice. At the same time, increased FGF21 could also affect hepatic lipid metabolism, as suggested by Fisher et al. We integrated these considerations in the introduction, results, and discussion sections in our revised manuscript. We also amended our model based on the revised mechanism in Fig 4d.

The 2015 review cited minimizes effects of FGF21 on the liver which have since been acknowledged and reviewed by others.

→ We discussed the current hypothesis regarding the role of FGF21 on the liver in the discussion section of our revised manuscript.

In listing transcription factors that regulate FGF21, ChREBP needs to be added. This has been reported by several, most recently by Fisher in molecular metabolism.

→ We added the role of ChREBP as a transcription factor for FGF21 in the introduction section as well as in the result section in our revised manuscript.

Reviewer #2 (Remarks to the Author):

In this revised study, with additional new data, the authors convincingly show that the prolonged CRTC2 activity, under insulin resistance, not only activates hepatic glucose production, but also promotes lipid accumulation by activation of a miR-34a-SIRT1/PPAR α -FGF21 axis. Overall, this is an excellent study that will advance our understanding of the metabolic function of CRTC2.

→ I sincerely appreciate the reviewer's comments regarding our manuscript.